# All Pedicle Screw versus Hybrid Hook–Screw Instrumentation in the Treatment of Thoracic Adolescent Idiopathic Scoliosis (AIS): A Prospective Comparative Cohort Study

**DOI:** 10.3390/healthcare10081455

**Published:** 2022-08-03

**Authors:** Athanasios I. Tsirikos, Tristan E. McMillan

**Affiliations:** 1Scottish National Spine Deformity Centre, Royal Hospital for Children and Young People, University of Edinburgh, 50 Little France Crescent, Edinburgh Bioquarter, Edinburgh EH16 4TJ, UK; 2Scottish National Spine Deformity Centre, Royal Hospital for Children and Young People, 50 Little France Crescent, Edinburgh Bioquarter, Edinburgh EH16 4TJ, UK; tristan.mcmillan@nhs.net

**Keywords:** scoliosis, adolescent idiopathic, thoracic, surgical correction, hybrid instrumentation, pedicle screw instrumentation, outcomes

## Abstract

Background: Posterior spinal correction and fusion remains the most common surgical treatment in AIS. Surgeons currently favour all pedicle screw (AS) correction techniques with alternative implants being less utilised. The purpose of this study was to assess whether a hybrid hook–screw (HS) construct could achieve similar outcomes. Methods: A single centre, prospective cohort study was conducted. Patients with moderate and severe thoracic AIS (Lenke 1) were included. Clinical and radiological results of a standardised hybrid HS technique were compared with those obtained with an AS construct. All patients had a minimum 2-year follow-up. Results: 160 patients were included in this series (80 patients/group). The HS group had significantly reduced surgical time, blood loss and implant density. Both techniques achieved ≥75% scoliosis correction. The HS group was superior in restoring thoracic kyphosis and global sagittal balance with an average 31% increase in kyphosis compared to 10% with the AS group (*p* < 0.001). There was significant improvement in SRS-22 scores at 2 years postoperative (*p* < 0.001) in both groups. There were no neurological or visceral complications related to instrumentation, no detected non-union and no reoperations. The HS implant cost was significantly lower than that of AS, with a mean instrumentation saving of almost £2000/patient. Conclusion: A standardised hybrid HS technique achieved excellent correction of thoracic scoliosis, high patient satisfaction and low complication rates in patients with thoracic AIS. These results were comparable to the AS group. The HS technique achieved better correction of thoracic kyphosis and sagittal balance than the AS technique, together with reduced surgical time, blood loss and implant cost.

## 1. Introduction

Surgical treatment in idiopathic scoliosis aims to restore global coronal and sagittal spinal balance, address rotational deformity and minimise complications [1]. Techniques have evolved in the last 60 years, particularly with the advent of pedicle screws. Cotrel et al. [2] introduced the Cotrel–Dubousset segmental instrumentation to achieve three-dimensional scoliosis correction. All-pedicle screw constructs were developed to obtain three-column fixation, segmental correction and vertebral derotation; this is currently the benchmark for adolescent idiopathic scoliosis (AIS) correction [3,4]. 

All-screw techniques can lead to a loss of thoracic kyphosis, particularly in high implant density constructs [5]. The resultant hypokyphosis can cause patient morbidity from adjacent level junctional deformity and accelerated degeneration due to global sagittal imbalance. Furthermore, pedicle screw insertion is not a benign procedure, with reported misplacement rates of 15.7% per thoracic screw when assessed with postoperative computed tomography (CT) [6]. Additionally, the use of segmental pedicle screws has resulted in a significant increase in the procedural cost of treating AIS [7]. To address some of these concerns in scoliosis surgery, it is possible to combine pedicle screw constructs with alternative implants that facilitate rod fixation to vertebrae in hybrid constructs. Such techniques include the use of pedicle hooks, sublaminar wires, bands or acrylic loops [8,9].

The purpose of this study was to assess clinical and radiological results of patients with thoracic AIS treated by a hybrid pedicle–hook–screw technique (HS) compared to segmental all-screw instrumentation (AS). Additionally, we compared patient reported outcomes and implant costs between these techniques. 

## 2. Materials and Methods

We performed a prospective cohort study including patients with thoracic AIS (Lenke type 1) treated by posterior spinal fusion, under the senior author (A.I.T.). Magnetic resonance imaging excluded intraspinal anomalies in all patients. We recorded patients’ gender, age and Risser grade at surgery. The vertebral levels fused, surgical time (skin incision to closure) and intra-operative blood loss [total volume and percentage of estimated blood volume (EBV)] were collated. Patients had a minimum 2-year postoperative follow-up beyond skeletal maturity (mean: 4.2 years; range: 2.5–6 years) with clinical, radiological and functional outcomes using the Scoliosis Research Society Outcomes Questionnaire (SRS-22) [10]. The SRS-22 data were collected preoperatively, at 6-, 12- and 24-months post-surgery, input into the British Spine Registry and analysed by an independent data co-ordinator. Local institutional review board approval was obtained for the study. 

### 2.1. Radiographic Evaluation 

All spinal radiographs were measured on a digital system (Carestream Health Ltd., Hemel Hempstead, Hertfordshire, UK) by both authors. Radiographs included posteroanterior and lateral erect, as well as preoperative supine traction views. The following measurements were taken before surgery and at latest follow-up: scoliosis angle, apical vertebra translation (AVT), lowest instrumented vertebra angle (LIVA), thoracic kyphosis (T1-T12), lumbar lordosis (L1-S1), global coronal balance and the distance between the lateral C7 plumb line and the postero-superior corner of S1 to assess global sagittal balance. We calculated the flexibility index (FI, %) [(preoperative Cobb angle–supine traction angle)/preoperative Cobb angle) × 100] and correction index (CI, %) [(preoperative Cobb angle–postoperative Cobb angle)/preoperative Cobb angle) × 100]. Shoulder symmetry was assessed with clavicle angle and shoulder height difference. Rib index (RI) was measured using the method described by Grivas to quantify the severity of double rib contour (DRC) and evaluate chest wall deformity [11]. 

### 2.2. Surgical Technique

All patients had IV cefuroxime at anaesthesia induction and two doses postoperatively. Multimodal spinal cord monitoring recorded MEPs, cortical and cervical SSEPs and EMGs. Soft tissue releases and facetectomies were performed to increase spinal flexibility and allow deformity correction, before progressing to spinal instrumentation using a dual rod construct without cross-linkage. 

A. *All-pedicle screw technique*. This utilised segmental polyaxial reduction screws that can lock into monoaxial screws (favoured angle screws) and titanium rods (Expedium 5.5 system; DePuy/Synthes Spine, Raynham, MA, USA). Pedicle screws were placed bilaterally across the two cephalad- and caudal-instrumented vertebrae to provide proximal and distal construct stability. Additional pedicle screws were used across the convexity of the thoracic scoliosis to allow segmental correction. Concave apical screws were not placed. Correction was achieved over the convex rod through a cantilever manoeuvre with segmental vertebral translation/derotation across the apical scoliotic levels (Figure 1) [12]. 

B. *Hybrid hook–screw technique.* This technique included a combination of pedicle hooks proximally and monoaxial pedicle screws distally using the Universal Spine System (USS II, DePuy/Synthes Spine, Raynham, MA, USA). Bilateral screws across the two lowest and pedicle hooks in the three upper instrumented vertebrae provided proximal and distal construct stability. Pedicle screws were inserted on the concave side, caudal to the apical vertebra. The scoliosis was corrected over the concave rod using rod derotation, which achieved translation of the apical vertebrae towards the midline (Figure 2). The concave rod was pre-contoured to approximately 60° thoracic kyphosis in order to restore sagittal balance of the spine as some of this kyphosis is lost during rod engagement to the pedicle screws and during rod derotation. The convex rod was loosely engaged to the proximal and distal fixation points during the concave correction manoeuvres to prevent exacerbation of rib prominence that can occur during en-bloc rod derotation. Proximal and distal distraction/compression was performed to achieve level shoulders and waist.

In both cohorts, pedicle screw placement was performed using a free hand technique based on anatomical landmarks [13]. Implant positioning and curve correction was assessed using an intra-operative image intensifier. This was followed by decortication of the posterior elements and onlay of bone grafts. Locally harvested autologous bone was supplemented by allograft to achieve fusion. All patients followed an enhanced recovery pathway in the ward. Postoperative trunk support was not used. 

### 2.3. Statistical Analysis

Data were analysed using IBM SPSS v. 27.0 (Armonk, NY, USA). A Shapiro–Wilk test was used to assess data normality. An independent samples t-test compared continuous parametric data between groups, with a Mann–Whitney test used for non-parametric data. Categorical binary data were analysed using the chi-squared test. Two-tailed *p*-values were reported, with significance at *p* < 0.05. Pearson’s and Spearman’s correlation coefficient measured the strength of the linear relationship between variables (r > 0.2 was considered significant).

### 2.4. Implant Cost Analysis

Spinal instrumentation cost was calculated for each patient and compared between the AS and HS techniques. 

## 3. Results

Table 1 summarises the demographic data. There were 80 patients per cohort who covered two different chronological periods of our practice (HS group: 80 consecutive patients operated between 2015–2018; AS group: 80 consecutive patients operated between 2008–2012) with female predominance (83% of whole group). There was no selection bias between the two groups as each group included 80 consecutive adolescent patients with the same type of scoliosis all operated during a specific chronological period under the care of the senior author. Mean age at surgery was 15.1 years in the HS group and 15.6 years in the AS group (*p* = 0.11). Mean Risser grade was 3 and 3.1, respectively (*p* = 0.68). All patients had thoracic scoliosis with a predominance of lumbar modifier Grade A [Centre Sacral Vertical Line (CSVL) lies between the pedicles of the stable vertebra; 88% of whole group]. There was no difference in the distribution of Lenke modifiers between cohorts (*p* = 0.32). 

There was no difference in the mean number of vertebrae fused between groups (Table 2). The HS group had an implant density of 1.1 compared to 1.4 in the AS group (*p* < 0.001). Surgical time and intra-operative blood loss were reduced in the HS group (*p* < 0.001). Increased surgical time correlated with an increased number of vertebrae fused (r = 0.33, *p* < 0.001), increased implant density (r = 0.63, *p* < 0.001) and greater blood loss (r = 0.65, *p* < 0.001). Mean hospital stay was 5 days (range: 3–9 days), with no difference between groups (*p* = 0.81).

### 3.1. Coronal Parameters

Mean preoperative thoracic scoliosis was 64° (range: 44–90°) for the AS and 62° (range: 42–94°) for the HS group (*p* = 0.31) (Table 3). At follow-up, mean thoracic scoliosis was 14° (range: 0–44°) for the AS and 16° (range: 2–40°) for the HS group. FI correlated with CI (r = 0.3, *p* < 0.001). Increased preoperative thoracic scoliosis correlated with lower FI (r = −0.4, *p* < 0.001) and greater postoperative scoliosis (r = 0.5, *p* < 0.001). Mean preoperative AVT was 6.6 cm (range: 3–13 cm) in the AS and 6.1 cm (range: 1–13 cm) in the HS group, improving to mean 1.5 cm (range: 0–4 cm) and 1.9 cm (range: 0–5 cm), respectively, at follow-up. AVT correction was better in the AS group (78% versus 70%, *p* = 0.001). Mean preoperative LIVA was 16.3° (range: 0–31°) in the AS and 22.7° (range: 5–40°) in the HS group. At follow-up this corrected to 3.1° (range: 0–13°) in the AS and 6.3° (range: 0–19°) in the HS group (*p* < 0.001).

Mean compensatory lumbar scoliosis was 37° (range: 25–44°) in the AS and 41° (range: 28–57°) in the HS group. This improved spontaneously to mean 14° (range: 6–23°) in the AS and 14° (range: 4–28°) in the HS group at follow-up after instrumented correction of the primary thoracic scoliosis. There was no difference in compensatory lumbar scoliosis before or after surgery between groups (*p* > 0.05).

In both groups, patients had preoperative coronal imbalance with a mean trunk shift of 1.4 cm (range: 0–5 cm) in the AS and 1.9 cm (range: 0–7 cm) in the HS group. Coronal imbalance correlated with the degree of scoliosis (r = 0.2, *p* = 0.01). At follow-up, the mean trunk shift improved to 0.2 cm in the AS and 0.1 cm in the HS group. Similarly, both groups demonstrated shoulder asymmetry, with a mean shoulder height difference of 1.6 cm (range: 0–6 cm) in the AS and 2 cm (range: 0–6 cm) in the HS group, and a mean clavicle angle difference of 3.5° (range: 0–12°) and 3.8° (range: 0–12°), respectively. In both groups, this asymmetry improved to a mean shoulder height difference of < 0.5 cm and a clavicle angle difference of < 1° at follow-up. Clavicle angle difference correlated with shoulder height difference (r = 0.9, *p* < 0.001). Scoliosis correction and implant density did not correlate with shoulder balance as expressed by postoperative clavicle angle or shoulder height difference.

### 3.2. Sagittal Parameters

Preoperatively, mean thoracic kyphosis was 38° (range: 3–80°) in the AS and 31° (range: −17–65°) in the HS group (*p* = 0.01). At follow-up, there was no difference in kyphosis between the groups (*p* = 0.2); however, kyphosis correction was greater in the HS group (*p* < 0.001). Thoracic kyphosis CI and thoracic kyphosis at follow-up inversely correlated with implant density (r = −0.2, *p* = 0.002). Similarly, global sagittal balance at follow-up was improved when an implant of a reduced density was used (r = −0.5, *p* < 0.001). This indicates that higher implant density limits the ability to restore thoracic kyphosis and global sagittal balance. Greater correction of thoracic scoliosis correlated with a postoperative decrease in thoracic kyphosis (r = −0.2, *p* = 0.02). This suggests that increased thoracic scoliosis correction was achieved at the expense of restoring thoracic kyphosis.

There was no difference between the groups when comparing lumbar lordosis before surgery and at latest follow-up or when comparing change in lordosis after surgery. Mean global sagittal balance was −2.1 cm in the AS and −1.1 cm in the HS group (*p* = 0.03). Sagittal balance improved after surgery in both groups, but there was better correction (86%) and mean postoperative sagittal balance (*p* < 0.001) in the HS group.

### 3.3. Axial Parameters

There was no difference in RI between the groups before surgery (*p* = 0.84). Increased preoperative RI correlated with greater preoperative scoliosis (r = 0.3, *p* < 0.001). At follow-up, RI was 1.46 in the HS and 1.6 in the AS group (*p* = 0.002), with RI = 1.00 indicating no rotational chest deformity. Scoliosis CI or implant density did not correlate with RI correction.

### 3.4. Complications

Superficial wound infection developed in three patients in each group (3/80, 3.8%). This resolved with dressing care and oral antibiotics. No patient required surgical debridement. There was no implant failure, detected non-union or reoperation among this cohort. We had no neurological or visceral complications and no intra-operative neuromonitoring events. At follow-up, nine patients (9/160, 5.6%) had proximal junctional kyphosis (PJK). This included four patients in the HS and five in the AS group, with mean PJK 13° (range: 10–16°) and 15° (range: 13–17°), respectively. PJK remained stable and all patients were asymptomatic with no need to extend the construct/fusion.

### 3.5. Clinical Outcomes

Both groups demonstrated improvement in SRS-22 total and domain scores between preoperative examination and 2-years post-surgery (*p* < 0.001). At a 2-year follow-up there was no difference in SRS-22 scores between the groups (4.36 versus 4.41, *p* = 0.52). There was no difference in SRS-22 total scores between the groups at the other collected time points (Table 4). The greatest improvement was seen in self-image. At a 6-month review, mean scores for function were reduced to below preoperative levels as we restricted physical activities to allow bone healing. SRS-22 for function improved at 1- and 2-year assessment points (Figure 3 and Figure 4). Higher preoperative SRS-22 scores correlated with improved scores at 6 (r = 0.49, *p* < 0.001), 12 (r = 0.45, *p* < 0.001), and 24 months (r = 0.49, *p* < 0.001) after surgery. SRS-22 scores and patient satisfaction did not correlate with any parameter of spinal deformity (coronal, sagittal and axial) or rib/shoulder asymmetry correction.

### 3.6. Cost Analysis

Mean implant cost was £6463.26 (range: £4279.50–8256.00) for the AS and £4535.49 (range: £2251.54–6331.73) for the HS technique (*p* < 0.001).

## 4. Discussion

The AS and HS groups presented in this study are matched with equal patient numbers and no difference in age, Risser grade, Lenke type and scoliosis angle, allowing for comparison. There was no selection bias between the two surgical techniques as the two cohorts covered different chronological periods of our practice. Radiographic and SRS outcomes from this series compared to previous studies are presented in Table 5. A mean scoliosis correction of 76% was achieved among this study cohort, which is comparable to other series [8,13,14,15]. Scoliosis correction correlated with the degree of preoperative scoliosis and FI; larger curves produced a stiffer spine and were corrected less. We recorded no difference between the two techniques for postoperative main thoracic and compensatory lumbar scoliosis. The difference in postoperative LIVA among groups is attributed to increased preoperative LIVA in the HS group. Although there was no difference in scoliosis correction, the AS group demonstrated better correction in AVT compared to the HS group. Similarly, whilst the HS group showed greater improvement in postoperative coronal balance, shoulder symmetry and clavicle angle, differences of <1 cm or <1° are not clinically significant.

The hybrid technique utilised in this study avoided apical and upper thoracic pedicle screws to reduce the neurological/vascular risks associated with screw placement. Concave apical screws can also flatten the concave rod during scoliosis correction, limiting the ability to restore thoracic kyphosis. Other authors have advocated using apical sublaminar/spinous process wires or hooks in a hybrid technique to facilitate scoliosis correction, although their clinical results were not superior to this study [8,17,21,22]. 

In the sagittal plane, both groups demonstrated improved thoracic kyphosis and global balance, but these parameters were superior when the HS technique was used. Quan et al. [23] showed a correlation between an increase in thoracic scoliosis correction and decrease in postoperative thoracic kyphosis, which was also seen in our results (r = −0.2, *p* = 0.02). This supports the benefit of accepting some residual scoliosis over a balanced spine, in order to better restore thoracic kyphosis. Greater implant density may reduce the ability to recreate thoracic kyphosis [5,24]. In this study, higher implant density correlated with reduced kyphosis and global sagittal balance correction. While implant density was low in both groups, it was reduced in the HS group, which achieved better correction of regional and global sagittal parameters compared to the AS group.

The development of convex rib prominence in thoracic AIS is a focal patient concern due to the impact on surface shape and body image. The correction of apical vertebral rotation may be improved by the three-column fixation of pedicle screws [25,26]. Asghar et al. [27] performed CT and reported a 22% correction of vertebral rotation with a hook–rod construct compared to 60% when AS instrumentation was used (*p* < 0.001). In this study, we evaluated rib hump correction with the DRC sign and calculated RI on lateral radiographs, not requiring CT scans that increase radiation exposure in young patients. Both groups achieved improvement in RI, which was greater when the HS technique was used. RI correction was not affected by scoliosis correction or implant density.

The HS group had reduced intra-operative blood loss and surgical time. This did not affect the length of hospital stay. We recorded low complication rates with both AS and HS techniques. Silvestre et al. [18] reported a higher incidence of surgery-related complications in the AS group compared to the HS group (44% versus 26%), including dural lesions and misplaced screws. Thoracic pedicle hooks may have lower risks of neurological or vascular injury due to malposition compared to pedicle screws, but this has not been shown in our study.

There was no difference in SRS-22 scores between the groups at 6-, 12- and 24-months post-surgery. Of all clinical and radiographic parameters, the only predictor of spinal-related health and patient satisfaction at 24 months was the SRS-22 score before surgery. Arlet et al. [20] reported no difference in postoperative SRS-24 scores between AS and HS constructs; there was no difference in cosmetic result, shoulder balance, trunk shift, rib hump and waist asymmetry when the two techniques were assessed by medical experts.

Larson et al. [28] reported that a reduction in the number of pedicle screws could decrease AIS treatment costs by up to 7%. On implant costs alone, we demonstrated a 30% saving using the HS technique, with AS constructs costing on average almost £2000 more per patient. Due to the shorter surgical time recorded when the HS technique was used, the overall patient cost reduction compared to the AS group was greater. 

This study has limitations. Firstly, whilst we compared implant costs, we did not perform a complete treatment cost analysis. Secondly, the AS group we used for comparison had a low implant density using a convex correction technique. Previous series using this technique showed comparable deformity correction and patient satisfaction to bilateral AS instrumentation [13,29]. In addition, Shen et al. [30] reported no difference in radiographic and clinical correction using low- versus high-density pedicle screw instrumentation in Lenke 1 AIS. 

In conclusion, in this study a standardised hybrid HS technique with low implant density achieved comparable correction of thoracic AIS to an AS construct, with high patient satisfaction and low complication rates. This was combined with improved correction of thoracic kyphosis and sagittal balance, as well as reduced surgical time and blood loss. These findings support the use of hybrid HS techniques in this select patient group. As global healthcare costs rise, cost savings with this technique are of increasing importance.

## Figures and Tables

**Figure 1 healthcare-10-01455-f001:**
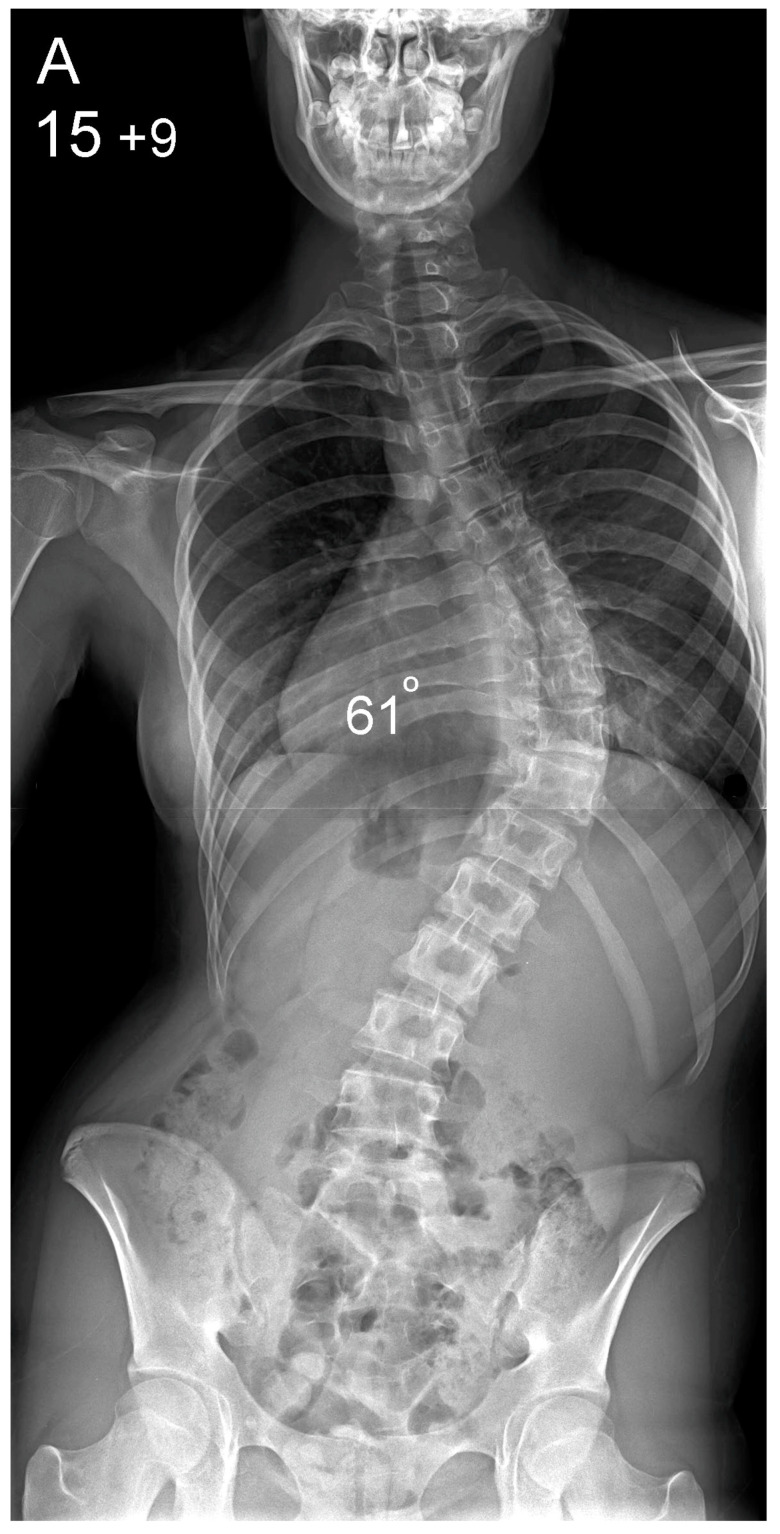
Patient aged 15 years and 9 months with a right thoracic AIS producing thoracic translocation and listing of the trunk to the right, as well as thoracic flat back producing negative global sagittal balance of the spine and compensatory cervical kyphosis (**A**,**B**). The patient underwent posterior scoliosis correction using the AS technique which restored segmental and global coronal/sagittal spinal balance at latest follow-up (age 18 years and 8 months) into adult life (**C**,**D**). Clinical photographs demonstrate excellent correction of the coronal deformity and associated rib hump after scoliosis surgery (**E**–**H**).

**Figure 2 healthcare-10-01455-f002:**
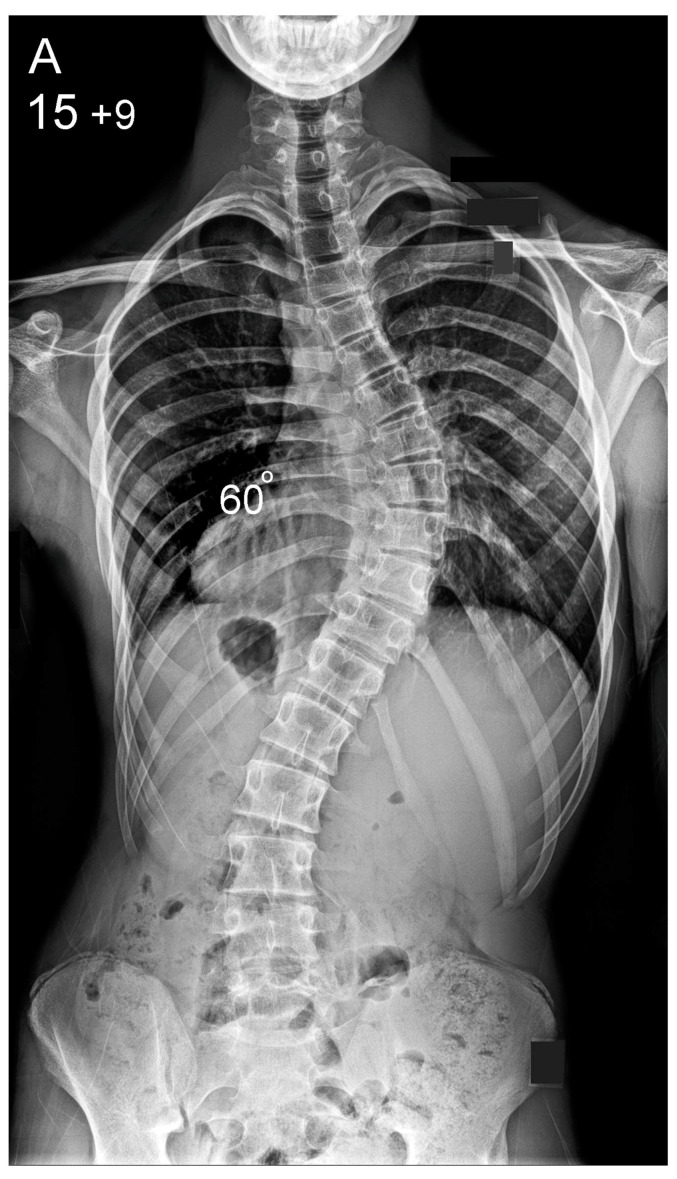
Patient aged 15 years and 9 months with a right thoracic AIS producing thoracic translocation and listing of the trunk to the right, as well as a lordotic thoracic spine causing spinal penetration into the chest, partial bronchial obstruction and right lower lobe atelectasis (**A**,**B**). The patient underwent posterior scoliosis correction using the HS technique which restored segmental and global coronal/sagittal spinal balance at latest follow-up (age 19 years and 2 months) into adult life (**C**,**D**). Clinical photographs demonstrate excellent correction of the coronal deformity and associated rib hump after scoliosis surgery (**E**–**H**).

**Figure 3 healthcare-10-01455-f003:**
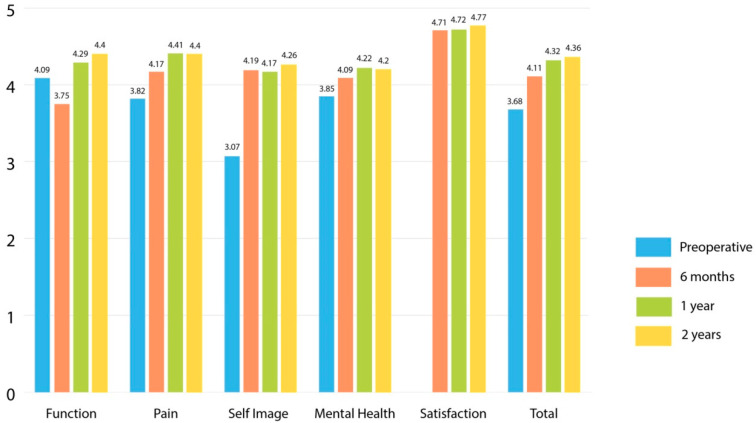
SRS-22 outcomes for patients treated with the all-pedicle screw (AS) technique.

**Figure 4 healthcare-10-01455-f004:**
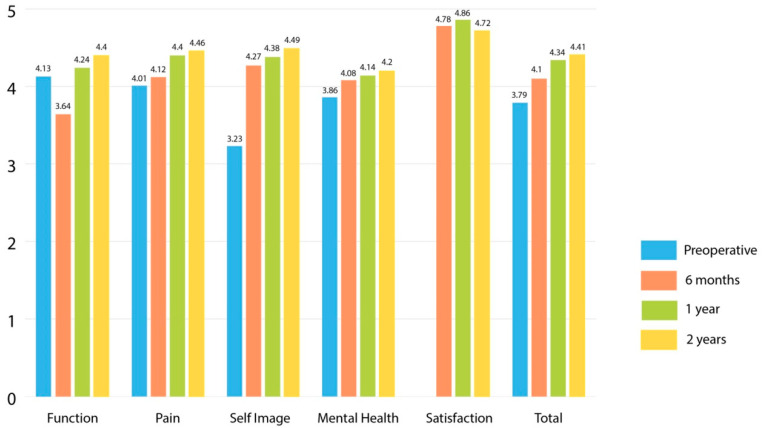
SRS-22 outcomes for patients treated with the hybrid hook–screw (HS) technique.

**Table 1 healthcare-10-01455-t001:** Patient demographics in the two groups.

Demographics	All-Screw (AS) Technique -No/Mean (Range)	Hybrid Hook–Screw (HS) Technique -No/Mean (Range)		*p*-Value
Patients	80	80		
Male	9	18		
Female	71	62		
Age at surgery (years)	15.6 (11.2–17.8)	15.1 (11.7–17.9)		0.13
Risser grade	3.1 (0–5)	3 (0–5)		0.68
Lenke curve type				
Lenke 1A	73	67	0.32	
Lenke 1B	6	10		
Lenke 1C	1	3		
Main thoracic scoliosis mean (range)	64° (44–90)	62° (42–94)		0.31

**Table 2 healthcare-10-01455-t002:** Surgical data in the two groups.

	All-Screw (AS) Technique -Mean (Range)	Hybrid Hook–Screw (HS) Technique -Mean (Range)	*p*-Value
Surgical time (min)	175 (120–240)	131 (90–180)	**<0.001 ***
Blood loss (mL)	900 (368–2000)	535 (129–2400)	**<0.001 ***
Blood loss (% of EBV)	23 (10–40)	13 (3–40)	**<0.001 ***
No. of vertebrae fused	11.4 (8–15)	11.3 (7–15)	0.73
Implant density	1.4 (1.2–2)	1.1 (0.7–1.3)	**<0.001 ***
Hospital stay (days)	5.4 (3–9)	5.3 (3–9)	0.81

* Statistically significant if *p* < 0.05 (in bold letters).

**Table 3 healthcare-10-01455-t003:** Comparison of the pre- and postoperative radiographic parameters between the two groups.

	All-Screw (AS) Technique -Mean (Range)	Hybrid Hook–Screw (HS) Technique -Mean (Range)	*p*-Value
**Main thoracic scoliosis**			
Preoperative (°)	64 (44–90)	62 (42–94)	0.31
Flexibility index (%)	34 (8–67)	37 (5–84)	0.11
Postoperative (°)	14 (0–44)	16 (2–40)	0.1
Correction Index (%)	78 (42–100)	75 (50–96)	0.07
**Apical Vertebral Translation**			
Preoperative (cm)	6.6 (3–13)	6.1 (1–13)	0.07
Postoperative (cm)	1.5 (0.4)	1.9 (0–5)	**0.02 ***
Correction Index (%)	78 (37–100)	70 (18–100)	**0.001 ***
**Lowest Instrumented Vertebra Angle**			
Preoperative (°)	16.3 (0–31)	22.7 (5–40)	**<0.001 ***
Postoperative (°)	3.1 (0–13)	6.3 (0–19)	**<0.001 ***
Correction Index (%)	79.6 (67–100)	73.1 (28–100)	**0.002 ***
**Compensatory lumbar scoliosis**			
Preoperative (°)	37 (25–44)	41 (28–57)	0.07
Postoperative (°)	14 (6–23)	14 (4–28)	0.23
Correction Index (%)	60 (38–81)	62 (4–88)	0.57
**Thoracic Kyphosis**			
Preoperative (°)	38 (3–80)	31 (−17–65)	**0.01 ***
Postoperative (°)	42 (16–62)	43 (30–58)	0.2
Correction Index (%)	10 (38–92)	31 (34–149)	**<0.001 ***
**Lumbar Lordosis**			
Preoperative (°)	55 (28–84)	53 (21–90)	0.33
Postoperative (°)	47 (25–68)	45 (29–62)	0.11
Correction Index (%)	18% (0–42)	20% (0–44)	0.64
**Coronal balance**			
Preoperative (cm)	1.4 (0–5)	1.9 (0–7)	**0.05**
Postoperative (cm)	0.2 (0–1.5)	0.1 (0–1)	**0.001 ***
Correction Index (%)	82 (0–100)	90 (0–100)	**0.01 ***
**Sagittal balance**			
Preoperative (cm)	−2.1 (−9.5–4)	−1.1 (−10.6–6)	**0.03 ***
Postoperative (cm)	−0.9 (−3–1)	−0.2 (−7–1)	**<0.001 ***
Correction Index (%)	57.4 (0–133)	86 (0–153)	**<0.001 ***
**Shoulder height difference**			
Preoperative (cm)	1.6 (0–6)	2 (0–6)	0.26
Postoperative (cm)	0.4 (0–2)	0.2 (0–1)	**0.02 ***
Correction Index (%)	61 (0–100)	80 (0–100)	**0.08**
**Clavicle angle difference**			
Preoperative (°)	3.5 (0–12)	3.8 (0–12)	0.26
Postoperative (°)	0.8 (0–4)	0.5 (0–3)	**0.02 ***
Correction Index (%)	63 (0–100)	74 (0–100)	0.18
**Rib Index**			
Preoperative	2.09 (1.4–3.7)	2.1 (1.5–3.2)	0.84
Postoperative	1.6 (1.1–2.4)	1.46 (1.1–2)	**0.002 ***
Correction Index (%)	23.4 (0–50)	30.5 (0–48)	0.09

* Statistically significant if *p* < 0.05.

**Table 4 healthcare-10-01455-t004:** Comparison of SRS-22 total scores preoperatively and at 6-, 12- and 24-month postoperative follow-up between the two groups.

	All-Screw (AS) Technique -Mean (Range)	Hybrid Hook–Screw (HS) Technique -Mean (Range)	*p*-Value
**Preoperative**	3.68 (2.18–4.5)	3.79 (2.15–4.65)	0.33
**6 months**	4.11 (2.14–5)	4.1 (2.27–4.91)	0.9
**12 months**	4.32 (2.77–5)	4.34 (3.23–4.95)	0.31
**24 months**	4.36 (2.86–5)	4.41 (2.91–5)	0.52

**Table 5 healthcare-10-01455-t005:** Comparison of radiological and SRS outcome questionnaire data between this study and previously published series [8,14,15,16,17,18,19,20].

	Liljenqvist et al., 2002 [14]		Kim et al., 2004 [15]		Lowenstein et al., 2007 [16]		Silvestre et al., 2008 [18]		Arlet et al., 2009 [20]		Luhmann el at., 2012 [19]			Yilmaz el at., 2012 [8]		Crawford et al., 2013 [17]		Current Study 2022	
**Type of correction technique/instrumentation**	AS	AH	AS	AH	AS	HS	AS	HS	AS	HS	AS	HS	AS	AH	HS	AS	HS	AS	HS
**Patient number**	50	49	26	26	17	17	25	27	20	20	53	48	35	35	35	34	29	80	80
**Major Cobb (°)**																			
Preoperative	63	61	63	66	55	47	88	92	49	52	52	57	59	56	56	51	50	64	62
Postoperative	28	30	16	33	15	18	40	51	14	13	19	26	16	29	21	15	14	14	16
**AVT (mm)**																			
Preoperative	45	50	51	55	N/R	N/R	60	71	N/R	N/R	48	50	52	47	50	44	41	66	61
Postoperative	21	18	16	28			31	36			15	18	17	28	21	12	9	15	19
**LIVA (°)**																			
Preoperative	21	18	23	22	N/R	N/R	24	20			20	20	20	19	22	N/R	N/R	16	23
Post operative	6	7	7	11			9	8	5	4	5	7	23	9	8	3	4	3	6
**Thoracic Kyphosis (°)**																			
Preoperative	30	22	31	27	30	26	35	35	N/R	N/R	23	29	22	21	18	18	22	38	31
Postoperative	28	26	17	22	19	22	28	32			27	34	12	22	22	18	17	42	43
**Lumbar Lordosis (°)**																			
Preoperative	46	45	61	64	44	41	46	44	N/R	N/R	57	60	57	61	58	43	48	55	53
Postoperative	45	46	55	59	36	35	41	41			56	62	59	60	59	37	36	47	45
**Coronal Balance (mm)**																			
Preoperative	15	11	16.3	18.0	10.0	11	11	14	N/R	N/R	2.0	2.1	1.7	1.7	1.5	11	13	14	19
Postoperative	7	8	10.1	14.3	6.1	6.7	4.4	7	3.8	3.0	1.1	1.4	0.9	13.1	9.8	21	15	2	1
**Sagittal Balance (mm)**																			
Preoperative	19	19	−7	−26	−29	−21	N/R	N/R	N/R	N/R	−22	−9	31	32	42	−53	−64	−21	−11
Postoperative	11	18	−20	−36	2	8			8	12	−35	−15	24	25	22	1	−38	−9	−2
**SRS-22 Overall Scores**			SRS 24:	SRS 24:			SRS 30:	SRS 30:	SRS 24	SRS 24	SRS 30	SRS 24	SRS 24	SRS 22	SRS 22			SRS 22:	SRS 22:
Preoperative	N/R	N/R	N/R	N/R	N/R	N/R	N/R	N/R								N/R	N/R	3.68	3.79
Postoperative			97	101			3.91	3.76	N/R	N/R	N/R	N/R	N/R	N/R	N/R			4.36	4.41

AS: All Screw, AH: All Hook, HS: Hybrid Hook Screw. N/R: Not recorded/listed in manuscript.

## Data Availability

All patients’ data including the patients’ reported outcomes (SRS-22 scores) were recorded in the British Spine Registry by the Data Coordinator of our Service.

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
