# Peer review of "All Pedicle Screw versus Hybrid Hook–Screw Instrumentation in the Treatment of Thoracic Adolescent Idiopathic Scoliosis (AIS): A Prospective Comparative Cohort Study"

_healthcare, 2022, doi:10.3390/healthcare10081455_

Round 1
Reviewer 1 Report
General impression
In this project, the authors assess clinical and radiological results of patients with thoracic adolescent idiopathic scoliosis (AIS) treated by hybrid pedicle hook-screw technique (HS) compared to segmental all-screw instrumentation (AI).
And they concluded that the HS technique achieved better correction of thoracic kyphosis and sagittal balance than the AS technique, together with reduced surgical time, blood loss and implant cost.
I believe the information in this study must be valuable for the physicians to perform surgery for AIS.
The methodology of this study was precisely explained and acceptable. Also, the limitations of this project were well indicated. For these reasons, I think this manuscript is appropriate for publication.
However, I have one minor request to be revised as stated below. After it has been resolved, I will judge this manuscript can be accepted and published by the healthcare journal.
1. page 16 paragraph 1 and 2
Units at these paragraphs were misprinted. That is, all units were indicated as “o”.
I evaluated this paper fairly and made judgement faithfully. I will be very pleased if you will respect my decision. However, I have one question. Postoperative mean thoracic kyphosis in the case series with AS technique was achieved just same as that in the case series with HS technique although correction index was significantly better in HS technique than in AS technique. Do the authors perform any special technique to obtain such results?Then, you should correct errors in the writings that I have indicated before publication. They should be corrected properly before publication.
Author Response
We would like to thank this Reviewer for taking the time to review our paper and for the constructive comments.
In response to these comments we would like to make the following points:
- We checked all units across our manuscript.
- Regarding our correction technique in the HS group, we pre-contoured the concave rod with increased thoracic kyphosis (approximately 60 degrees) as the concave rod tends to partly 'flatten' during coronal correction with the aim to restore sagittal balance of the spine. This allowed adequate restoration of thoracic kyphosis which was achieved parallel to the scoliosis correction. This point has been explained in the revised manuscript.
Reviewer 2 Report
This manuscript determines a new surgical approach that allow the restoration of scoliosis with our sacrificing the kyphosis in adolescents who undergone surgery for scoliosis correction. It is well written, with a good English and grammar level. To improve it, some issues are given to authors:
Reference style is wrong in the whole manuscript. I recommend authors to follow journal rules.
Authors must explain how participants were selected to be treated by one surgical technique or the other one. This is not clear and could highly influence in the results and the baseline comparison. Not solving this issue would require to include it in limitations and to smooth the conclusions.
Author Response
We would like to thank this Reviewer for taking the time to review our paper and for the constructive comments.
In response to these comments we would like to make the following points:
- We reviewed the guidelines of the Journal and have revised the reference style in our paper according to these.
- There was no selection bias for choosing patients who were part of the 2 groups, namely the HS and the AS groups. Eighty patients were included in each group and these were consecutive and were operated in different chronological periods of the practice of the senior author. These points have been clarified in the section under 'Results'.
Reviewer 3 Report
Dear author,
Thank you for the opportunity to review this article.
It seems to be a complex work that compares all-screw correction with hybrid screw-laminar technique regarding patients outcome and economical criteria.
It is a very thorough work with Materials and Methods exposed clearly and with a list of Results worthy of this field of interest.
Although you mentioned other studies that had different results, You seem to avoid selection bias by matching the same criteria among the study groups.
Can you elaborate on the correction technique of lateral translation?
Why did you apply the SRS-22 score instead the more elaborate SRS-30? Here is an article about "Quality of Life Evaluation Using SRS-30 Score for Operated
Children and Adolescent Idiopathic Scoliosis" published in Children, DOI https://doi.org/10.3390/
medicina58050674 that gives an insight upon the advantages on the SRS-30, for reference.
Overall, the paper meets the criteria for publishing after a revision.
Author Response
We would like to thank this Reviewer for taking the time to review our paper and for the constructive comments.
In response to these comments we would like to make the following points:
- There was no selection bias for choosing patients who were part of the 2 groups, namely the HS and the AS groups. Eighty patients were included in each group and these were consecutive and were operated in different chronological periods of the practice of the senior author. These points have been clarified in the section under 'Results'.
- Regarding our correction technique in the HS group, derotation of the concave rod has achieved translation of the apical vertebrae towards the midline as we used concave pedicle screw fixation from the most caudal vertebra up to the vertebra just below the concave apex of scoliosis. This allowed adequate coronal correction without the need for placing pedicle screws at the concave apical vertebra or the vertebrae above which carry increased risk of misplacement.
- We appreciate the Reviewer's comment regarding the use of the SRS-30 as a tool to evaluate quality of life in scoliosis patients. We used the SRS-22 tool in our study as this has been the patient assessment instrument that has been recommended by the British Spine Registry which is the National Database where we record all operated patients in our Service. The SRS-22 instrument has been widely used in the literature but also in our previous studies which allows comparison of the results of the current series.
Round 2
Reviewer 3 Report
The article could be publish in the present form.